

# ToRQuEMaDA: tool for retrieving queried Eubacteria, metadata and dereplicating assemblies

Raphaël R. Léonard[1,2], Marie Leleu[2,3], Mick Van Vlierberghe[2], Luc Cornet[2,4], Frédéric Kerff[1] and Denis Baurain[2]

[1] InBioS – Centre d'Ingénierie des Protéines, Université de Liège, Liège, Belgium
[2] InBioS –PhytoSYSTEMS, Eukaryotic Phylogenomics, Université de Liège, Liège, Belgium
[3] UGSF –Unité de Glycobiologie Structurale et Fonctionnelle, Université de Lille/CNRS, Lille, France
[4] Mycology and Aerobiology, Sciensano, Service Public Fédéral, Bruxelles, Belgium

## ABSTRACT

TQMD is a tool for high-performance computing clusters which downloads, stores and produces lists of dereplicated prokaryotic genomes. It has been developed to counter the ever-growing number of prokaryotic genomes and their uneven taxonomic distribution. It is based on word-based alignment-free methods ($k$-mers), an iterative single-linkage approach and a divide-and-conquer strategy to remain both efficient and scalable. We studied the performance of TQMD by verifying the influence of its parameters and heuristics on the clustering outcome. We further compared TQMD to two other dereplication tools (dRep and Assembly-Dereplicator). Our results showed that TQMD is primarily optimized to dereplicate at higher taxonomic levels (phylum/class), as opposed to the other dereplication tools, but also works at lower taxonomic levels (species/strain) like the other dereplication tools. TQMD is available from source and as a Singularity container at [https://bitbucket.org/phylogeno/tqmd].

Corresponding author
Denis Baurain,
denis.baurain@uliege.be

## INTRODUCTION

The fast-growing number of available prokaryotic genomes, along with their uneven taxonomic distribution, is a problem when trying to assemble high-quality yet broadly sampled genome sets for phylogenomics and comparative genomics. Indeed, most of the new genomes belong to the same subset of hyper-sampled phyla, such as Proteobacteria and Firmicutes, or even to single species, such as *Escherichia coli* (e.g., 105,081 out of 939,798 genomes in GenBank as of January 2021), while the continuous flow of newly discovered phyla prompts for regular updates of in-house databases. This situation makes it difficult to maintain sets of representative genomes combining lesser known phyla, for which only few species are available, and sound subsets of highly abundant phyla. An automated straightforward method is required but would be far too slow if based on regular alignment algorithms.

Alignment-free methods are quantifiable ways of comparing the similarity of sequences without using an alignment (*Zielezinski et al., 2017*). They have several advantages over alignment-based methods: they are computationally less expensive, they are resistant to gene shuffling and recombination events, and they do not depend on assumptions about sequence changes. In the review of (*Zielezinski et al., 2017*), two main categories of methods are described: the information theory-based methods and the word-based methods. The rationale behind word-based methods is that similar sequences share a similar set of words. Sequence words are called $k$-mers and can be defined as all the words, of a given size $k$, that one can enumerate for a given alphabet. The idea is to compare the "dictionaries" of the words observed in two different genomes. The more similar two genomes are, the more words their respective "dictionaries" will have in common. In contrast, information theory-based methods compute the amount of information shared between two analyzed (genomic) sequences. Several different ways to assess this quantity do exist (e.g., through data compression) but they are not the subject of this paper (see *Shannon, 1948*; *Kullback & Leibler, 1951*; *Kolmogorov, 1965*; *Tribus & McIrvine, 1971*; *Batista et al., 2011*; *Zielezinski et al., 2017* for details).

Based on the review on the alignment-free sequence comparison methods of (*Zielezinski et al., 2017*), two main categories of software packages were theoretically suitable for dereplicating prokaryotic genomes: the species identification/taxonomic profiling programs (Table 1 in *Zielezinski et al., 2017*) and the whole-genome phylogeny programs (Table 2 in *Zielezinski et al., 2017*). First, we did not investigate software solutions made available as web services because of their intrinsic limitation with respect to the amount of genomic data that one regular user can process through these interfaces. Second, all the programs belonging to the taxonomic profiling category required a reference database to compare the genomes to, which would have led us to a circular conundrum, in which a (possibly handmade) database of reference genomes is required to (automatically) build a database of representative genomes. Third, all those presented in the whole-genome phylogeny category were either not suited for large-scale dereplication or did not provide small enough running time estimates for their test cases. For example, jD2Stat (*Chan et al., 2014*) gives results for 5000 sequences of 1500 nucleotides in 14 min, which would clearly make computationally intractable the dereplication of hundreds of thousands of whole prokaryotic genomes. As of January 2021, we only found two programs that were made to dereplicate genomes, dRep (*Olm et al., 2017*) and Assembly-Dereplicator (*Wick & Holt, 2019*) . These two programs are presented below.

Considering the limitations of the existing tools for assembling representative sets of prokaryotic genomes, the present article describes our own program called "ToRQuEMaDA" (abbreviated TQMD in the following for convenience) for Tool for Retrieving Queried Eubacteria, Metadata and Dereplicating Assemblies. TQMD is a word-based alignment-free dereplicating tool for both public and private prokaryotic genomes designed for both high-performance computing (HPC) clusters and powerful single-node computing servers. TQMD is available on Bitbucket and can be installed on any HPC with SGE/OGE (Sun/Open Grid Engine) installed as a scheduler. Few modifications are needed to adapt the scripts to most local setups. A Singularity (*Kurtzer, Sochat & Bauer, 2017*)

container is also available for single-node computers without a scheduler. TQMD works both in parallel and iteratively. Using default parameter values, each elemental job takes two to three hours to complete (see Materials and Methods for test hardware specifications), and if enough CPUs are available to run all jobs of a given round at the same time, such a round should only take two to three hours. Usually, four to five rounds are sufficient to achieve the dereplication. Therefore, a single run of TQMD against ~60,000 Bacteria in NCBI RefSeq takes 8 to 15 h to complete.

## MATERIALS AND METHODS

### Hardware

Almost all the computational work was performed on a grid computer IBM/Lenovo Flex System composed of one big computing node (x440) and nine smaller computing nodes (x240), featuring a total of 196 physical cores, 2.5 TB of RAM and 160 TB of shared mass storage, and operating under CentOS 6.6. Beyond "bignode" (running the scheduler and the MySQL database; see below), only four of the smaller computing nodes were used when testing TQMD; their specifications are as follow: 2 CPUs Intel Xeon E5-2670 (8 cores at 2.6 GHz with Hyper-Threading enabled), 128 GB of RAM. For the dRep test (see below), we had to use a desktop workstation (Ubuntu Linux 16.04) featuring 2 CPUs Intel Xeon E5-2620 v4 (8 cores at 2.1 GHz with Hyper-Threading enabled) and 64 GB of RAM. Based on the comparator found on the website http://cpubenchmark.net/, the CPUs in our cluster and in the workstation were roughly equivalent (from −0.5 to +5% difference).

It is important to mention that due to Hyper-Threading configuration of the grid computer and the fact that several teams shared the infrastructure, queueing time and disk usage could not be strictly controlled during the tests. Therefore, all running times provided in this article are informed estimates rather than exact measurements. These estimates are those we would communicate to a user inquiring about the waiting time for a specific analysis to complete. They are an approximation of the running time recorded when the grid computer usage is low (i.e., almost no other user).

### Software architecture

TQMD is composed of a database and includes two main phases: (1) a periodic preparation phase in which newly available genome assemblies ("genomes" for short) are downloaded (or locally imported for private genomes) and individual genome metrics are computed, and (2) an "on-demand" dereplicating phase in which genomes (both new and old) are dereplicated on the fly to provide a list of high-quality representative genomes as a result (Fig. 1). The database stores the paths to the individual genome (FASTA) files, the individual genome metrics and the list of representative genomes produced by each TQMD run. Each piece of data is computed independently; if a dereplication request is issued during the computation of newly available genomes, TQMD only uses the genomes for which all the data is available in the database. Moreover, it is fully aware of the organisms (NCBI) taxonomy (*Federhen, 2012*), which means that taxonomic filters can be applied when downloading and/or when clustering genomes to spare time and/or focus on taxa of interest.

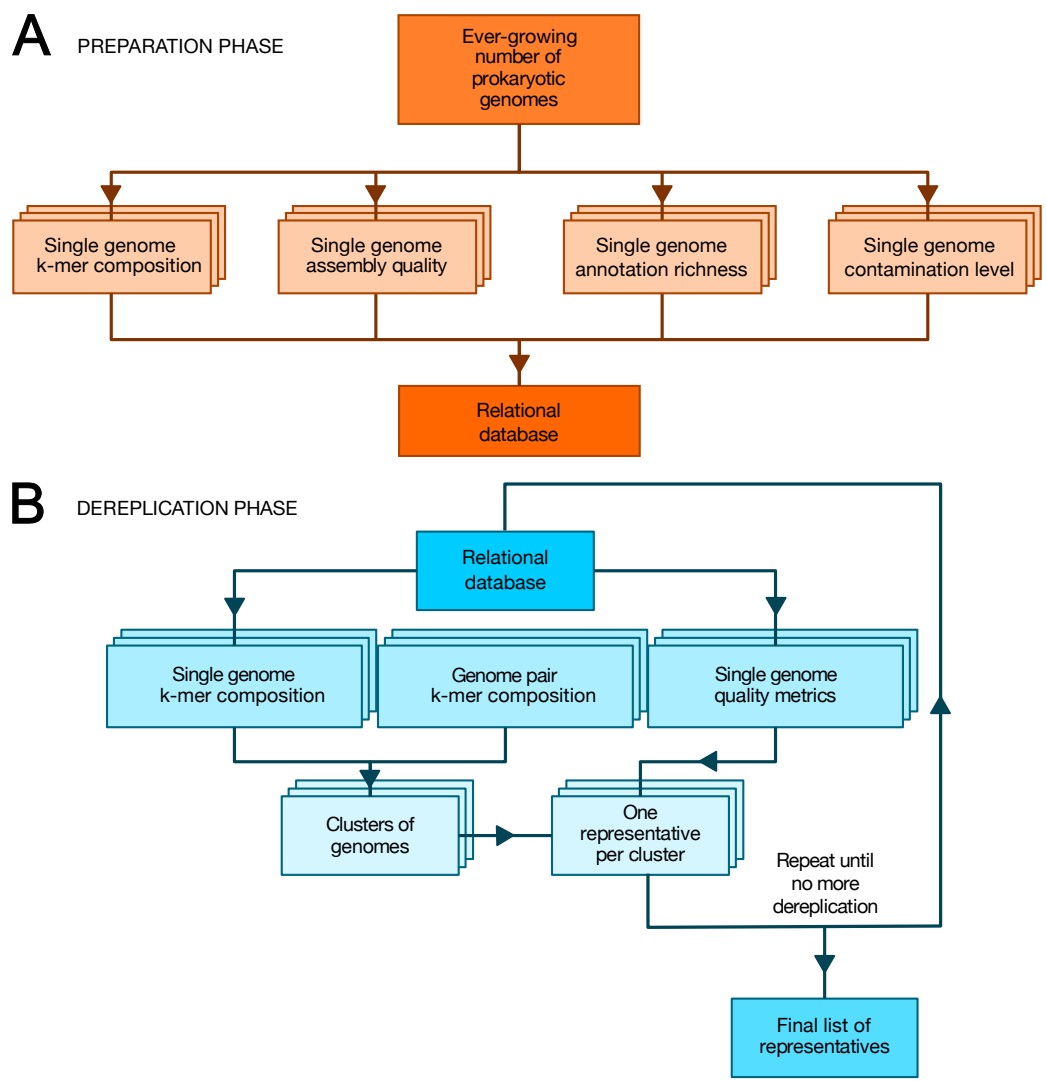

**Figure 1** **Overview of TQMD phases and heuristics.** (A) The preparation phase consists in download-ing newly released prokaryotic genomes from NCBI RefSeq and to pre-compute all per-genome informa-tion required to run the second phase: k-mer composition, assembly quality, annotation richness, con-tamination level (and completeness) level. Pre-computed information for single genomes is stored in a re-lational database associated with TQMD. (B) The dereplication phase then retrieves this information for all genomes to dereplicate from the database and clusters the genomes from pairwise distances computed on the fly. Cluster representatives (one per cluster created) are chosen for each cluster based on the single-genome metrics computed during the preparation phase. The dereplication is iterative and the process re-peats until representative genomes cannot be dereplicated anymore, which produces the final list of repre-sentatives. Parallelized steps are shown as overlaid boxes.

During the preparation phase, we download the genomes and proteomes and pre-compute all the data required by the dereplication phase to store them in the database: indexes of nucleotide $k$-mers for single genomes, genome assembly quality metrics, genome annotation richness metrics, Small Subunit ribosomal RNA (SSU (16S) rRNA) predictions, contamination level and completeness level, whereas during the dereplication phase, we

cluster the genomes based on these $k$-mer indexes and select a representative for each cluster based on a user-modifiable ranking formula taking into account assembly quality, annotation richness, contamination and completeness level. These criteria were chosen, so as to select the best representative genomes (*Bowers et al., 2017*). By that, we mean that representative genomes (if available) are expected to be fully sequenced, correctly assembled, richly annotated and devoid of contaminating sequences. To satisfy this last requirement, TQMD can also use optional contamination statistics produced by Forty-Two (*Irisarri et al., 2017*; *Simion et al., 2017*) and/or CheckM (*Parks et al., 2015*) (see below).

### Preparation phase

As shown in Fig. 1A, we first download the prokaryotic genomes from NCBI RefSeq (*O'Leary et al., 2016*) (or from GenBank (*Sayers et al., 2020*)). For the sake of data traceability, TQMD never gets rid of older genomes; newly released genomes are simply added to its internal database. The genomes from RefSeq and GenBank are kept physically separate. As TQMD was developed over five years, we have progressively accumulated several different versions of the RefSeq database, starting with release 79 (85,465 prokaryotic genomes, including 713 Archaea), then 79+92 (126,959 prokaryotic genomes, including 1,037 Archaea) and finally 79+92+203 (223,785 prokaryotic genomes, including 1,312 Archaea). Once RefSeq is up to date locally, we compute single-genome $k$-mer indexes and other metrics. For each of these computations, we use third-party programs and scripts (JELLYFISH, QUAST, RNAmmer, CD-HIT and Forty-Two or CheckM), except for the richness of the annotations, which we evaluate using an in-house script.

JELLYFISH (v1.1.12) (*Marçais & Kingsford, 2011*) is used to compute the $k$-mer indexes for single genomes (TQMD can also work with JELLYFISH v2.x and Mash (*Ondov et al., 2016*); see below). We tested several sizes for our $k$-mers. While JELLYFISH v1.x used to crash when using a size below 11 nucleotides, thus setting a hard lower bound on $k$-mer size, it is no longer an issue in JELLYFISH v2.x. On the other hand, while longer k-mers improve the specificity, they also require longer computing times (*Zielezinski et al., 2017*). With a size of 11, there are almost 4.2 millions ($4^{11}$) possible words. Consequently, a hypothetical genome featuring every possible $k$-mer without any repetition, could only be 4.2 Mbp long. Even if real genomes include repeats, genomes over 4 Mbp might still feature almost every $k$-mer, which would lead to useless $k$-mer indexes. To verify this idea, we examined the 85,465 genomes of RefSeq 79 and observed that about 15 genomes indeed almost exhaust the $k$-mer index (3 to 4 millions out of 4.2 millions), thus confirming that 11 is not a usable $k$-mer size. The next $k$-mer size, 12 nucleotides, offers over 16 millions ($4^{12}$) possibilities. The genomes with the largest index only reach 7.5 millions different $k$-mers, while the average index is below 2.7 millions $k$-mers. We could have used a $k$-mer size of 13 nucleotides, but our preliminary tests showed an important increase of the computing time. Whereas our tests with a $k$-mer size of 12 on all available Bacteria lasted between 8 and 15 h, depending on the distance threshold used (see below), our tests with a size of 13 required between 1 and 2 days to finish. Therefore, we chose to work with a $k$-mer size of 12 nucleotides. Above that, we would only have dereplicated closely related strains (i.e., belonging to the same species) due to a too high specificity (*Zielezinski et al., 2017*) and/or

the computing times would have become prohibitively long. Moreover, we did not use the "canonical" option for computing "strand-insensitive" $k$-mers with JELLYFISH (meant to be used on reads according to the manual) because we used RefSeq where the genomes are supposed to be fully assembled and thus gene orientation might be informative. If GenBank is used instead of RefSeq, it is highly recommended to enable the canonical option in TQMD due to the presence of genomes likely to be not assembled (still at the scaffold stage) or only very poorly assembled. Yet, one has to remember that canonical $k$-mers are twice less numerous for a given $k$-mer size than strand-specific $k$-mers, which might become an issue for distinguishing large genomes.

QUAST (v4.4) (*Gurevich et al., 2013*) is used to estimate the quality of genome assemblies (QUAST v5.x is also supported). We retrieve several quality metrics (13 in total) for each genome, these being the number of DNA sequences, the number of DNA sequences (or contigs) > 1 kbp, the size of the complete genome, the size of the complete genome composed of DNA sequences > 1 kbp, the number of contigs, the largest DNA sequence, the size of the complete genome composed of DNA sequences > 500 bp, the GC content, the N50, N75, L50 and L75 values, and the number of "N" per 100 kbp (N is the symbol used to scaffold contigs without matching ends). Given a minimal set of contigs ordered by descending length, the N50/N75 is defined as the length of the contig located at 50%/75% of the total genome length in the distribution, whereas the L50/L75 is defined as the rank of this specific contig. Among these metrics, we eventually decided to take into account (1) the relative length of the largest DNA sequence to the complete genome (> 1 kpb only) and (2) the fraction of "N" in the genome. In addition, we also use a size range (between 100 kbp and 15 Mbp) to remove the genomes too small to be complete and those too large to be considered uncontaminated (*Cornet et al., 2018b*).

For the richness of annotation, we compute what we call the "certainty" and the "completeness" of each genome. Importantly, this step necessitates (predicted) proteomes. While it is not an issue with RefSeq genomes, for which such predictions are always available, if TQMD is provided with an input genome set from a different source (GenBank or private genomes) with missing predicted proteomes, the related genomes will be automatically discarded (at least if the annotation metrics are used in the ranking formula). Our "certainty" metric corresponds to the proportion of sequences in a given proteome that we deem uncertain. To this end, we first count the number of sequence descriptions (in FASTA definition lines) with words indicating uncertainty, such as "probable", "hypothetical" or "unknown", then we compute a relative score as follows:

$$\text{Certainty} = 1 - \frac{\text{count of uncertain proteins}}{\text{total count of proteins}}$$

For "completeness", instead of counting the number of uncertain proteins, we count the number of proteins without any description:

$$\text{Completeness} = 1 - \frac{\text{count of unannotated proteins}}{\text{total count of proteins}}$$

Regarding genome contamination, RNAmmer (v1.2) (*Lagesen et al., 2007*) is used to predict the SSU (16S) rRNA of the genomes. By using cd-hit-est (v4.6) (*Li & Godzik, 2006*;

*Fu et al., 2012*) with an identity threshold of 97.5% (*Taton et al., 2003*), TQMD optionally creates a list of genomes featuring at least two SSU (16S) rRNA sequences belonging to different species (i.e., clustered in distinct CD-HIT clusters). This list of likely chimerical (or at least contaminated) genomes can be provided to filter out the genomes given as input to TQMD or produced in output by TQMD (see below). Another (more recent) possible threshold for species delineation based on SSU (16S) rRNA identity would be 99% (*Edgar, 2018*) and TQMD also supports such a setting.

Finally, another contamination metric is also available for the ranking: the genome contamination level estimated by the program Forty-Two (*Van Vlierberghe, 2021*) (v0.210570 or higher "42") based on the comparison of the genome ribosomal proteins to the reference sequences of the RiboDB database (*Jauffrit et al., 2016*). While this is the recommended approach for probing genome-wide contamination due to its speed, TQMD also supports CheckM (*Parks et al., 2015*) (v1.1.3) to predict "genome completeness" and "genome contamination". The contamination assessment of the latter is based on lineage-specific marker genes in addition to ribosomal proteins.

Once all these individual $k$-mer indexes and metrics are computed for all individual genomes, the genomes are ranked in a global ranking from the best to the worst genome (to be selected as a cluster representative), using an equal-weight sum-of-ranks approach available in the Perl module Statistics::Data::Rank. For each metric, a ranking is produced across all genomes and the final rank of a specific genome is computed as the sum of each of those individuals ranks without favoring one metric over another. For now, we do not consider all the metrics stored in the TQMD database, since all are optional and some are redundant. The five metrics (in TQMD syntax) used to compute the default ranking are: (1) assembly quality: quast.N.per.100.kbp; (2) assembly quality: quast.largest.contig.ratio (= quast.largest.contig / quast.total.len.1000.bp); (3) annotation richness: annot.certainty; (4) contamination level: 42.contam.perc; (5) contamination level: 42.added.ali. The first two metrics are obtained from QUAST, the third from our in-house script, and the fourth and fifth from "42". Finally, it is worth noting that TQMD allows the user to devise a custom ranking formula involving any combination of the 30 supported metrics (see details in TQMD manual).

### Dereplication phase

Genome clustering can be carried out on the full set of genomes stored in the TQMD database or only on one or more taxonomic subsets of them. Moreover, both positive (inclusion and/or representativeness priority) and negative (exclusion) lists of GCA/GCF numbers can be provided to alter TQMD input and output genome sets. TQMD itself optionally produces such a negative list to exclude genomes featuring multiple SSU (16S) rRNA sequences (see above). Furthermore, both public (from RefSeq/GenBank) and private (i.e., custom) genomes can be dereplicated simultaneously. Moreover, the presence of at least one SSU (16S) rRNA predicted by RNAmmer can be used as a requirement for the genome to be selected, which would rule out some metagenome-assembled genomes (MAGs), for which rRNA genes are often missing (*Cornet et al., 2018a*). Consequently, this option is recommended when working with RefSeq but not GenBank, at least if the selection

of some lesser quality MAGs is important for the user. Regarding priority lists, they can be useful in comparative genomics, when one wants to include model organism genomes without sacrificing dereplication. As shown in Fig. 1B, the dereplication process is iterative and stops once it deems itself finished. Its decision is based on three different convergence criteria, for which we provide default threshold values but these can be modified by the user (see below). TQMD stops cycling as soon as one criterion is satisfied.

Two different distances can be used for clustering genomes with TQMD, each one derived from a distinct similarity metric, the Jaccard Index (JI; see (*Real & Vargas, 1996*)) or the Identical Genome Fraction (IGF; see (*Cornet et al., 2018b*)), both applied to shared $k$-mers at the nucleotide level. The effective distance used by TQMD is then obtained by subtracting the corresponding similarity metric from 1.

The JI is a measure of the similarity between two finite datasets. It is defined as the intersection over the union of the two datasets A and B:

$$JI(A,B) = \frac{|A \cap B|}{|A \cup B|}$$

The JI can be computed in two different manners: (1) exact computation using JELLYFISH (default option) and (2) approximate estimation using Mash (*Ondov et al., 2016*) (v1.1.1). If Mash is to be used, precomputation of single-genome $k$-mers is not required.

The IGF, for Identical Genome Fraction, replaces the union in the JI by the size of the smallest of the two datasets A and B:

$$IGF(A,B) = \frac{|A \cap B|}{\min(A,B)}$$

The TQMD algorithm works similarly for both distances and is inspired by the greedy clustering approach implemented in packages such as CD-HIT (*Jones, Pevzner & Pevzner, 2004*; *Li & Godzik, 2006*; *Fu et al., 2012*). The greedy clustering can work in two different modes, loose and strict. In both cases, we first sort the list of genomes based on the global ranking of the genomes (assembly quality and annotation richness metrics, indicators of genome contamination; see above for details) and the top-ranking genome is assigned to a first cluster. Then, in loose mode, every other genome is compared to every member of every cluster until it finds a suitable cluster of similar genomes; otherwise, such a genome becomes the first member of a new cluster. Hence, the second genome is compared to the single genome of the first cluster. If its distance to the latter genome is lower than specified threshold (let us say it is the case here), it is added to the cluster. Similarly, the third genome is compared to the first member of the first cluster; if its distance is higher than the threshold, it is compared to the second member of this first cluster. If it still is higher than the threshold, and since there is no other cluster, it creates a new (second) cluster. The fourth genome follows the same path, as will all remaining genomes do until every genome of the list is assigned to a cluster, whether singleton or part of a larger group. As genomes are processed from the best to the worst in terms of global ranking, representative genomes (which correspond to cluster founders) are automatically the best possible for each cluster. In strict mode, every other genome is only compared to the representative genome (here

too corresponding to the highest-ranking genome) of every cluster, which both speeds up the clustering process and mitigates the potential drawbacks of pure single-linkage.

To scale up the greedy clustering algorithm, we used a divide-and-conquer approach (*Bentley, 1980*; *Jones, Pevzner & Pevzner, 2004*) (Fig. 2). Indeed, when performing our own tests, we worked with about 112,000 genomes, a number making clearly impossible to compare all genomes at once. Therefore, we first partitioned the list of genomes into smaller batches (hereafter termed "packs") of 200 by default, either based on their advertised (NCBI) taxonomy (*Federhen, 2012*; *Sayers et al., 2020*) or completely at random. The clustering of each small pack yields a single representative, which we regroup into a new (shorter) list of genomes that is processed iteratively following the same algorithm. In the next round, only the selected representatives are compared between each other, thereby precluding the genomes that were not selected to be directly compared. While this heuristic results in an important speed-up, it may also prevent similar genomes to be mutually dereplicated because they were processed in distinct packs and replaced by representatives that are potentially less similar. The iterative algorithm stops based on any of the following three criteria (which can be specified by the user): (1) if it reaches a maximum number of rounds, (2) if it falls below an upper limit for the number of representatives (i.e., number of clusters) or (3) if the clustering ratio between two successive rounds falls below a minimum threshold. We define the clustering ratio as the percentage of genomes dereplicated at the end of a TQMD round compared to the number of genomes still in the game at the beginning of the round.

## Phylogenomic analyses

We used TQMD runs as a source of representative bacterial genomes and obtained selections containing between 20 and 50 organisms for the six most populated phyla (the upper limit for the number of representatives was set to 50). We also generated two other selections to sample all Bacteria at once, one containing 49 organisms and the other 151. A last selection of Archaea was also produced and contained 86 organisms. For each TQMD run, we retrieved the proteomes of the selected representatives and used Forty-Two to retrieve their ribosomal proteins. Those proteins were taxonomically labelled by computing the last common ancestor of their closest relatives (best BLAST hits) in the corresponding alignments (excluding self-matches), provided they had a bit-score ≥80 and were within 99% of the bit-score of the first hit (MEGAN-like algorithm (*Cornet et al., 2018b*)). Thus, this strategy allowed us to simultaneously assess the completeness and the contamination level of each representative proteome while providing widely sampled ribosomal proteins for phylogenomic analyses (Table 1). For the bacterial dataset (B), the largest of the nine TQMD selections, this step took less than three hours to complete.

For each TQMD run, we assembled a supermatrix from the ribosomal proteins retrieved earlier (Table 1). Briefly, sequences were aligned with MAFFT v7.453 (*Katoh & Standley, 2013*), then the alignments were cleaned using ali2phylip.pl from the Bio::MUST::Core software package (D. Baurain, https://metacpan.org/release/Bio-MUST-Core), which implements the BMGE (*Criscuolo & Gribaldo, 2010*) filter (min=0.3, max=0.5, bmge-mask=loose). This step reduced the proportion of missing sites in the alignments. Next,

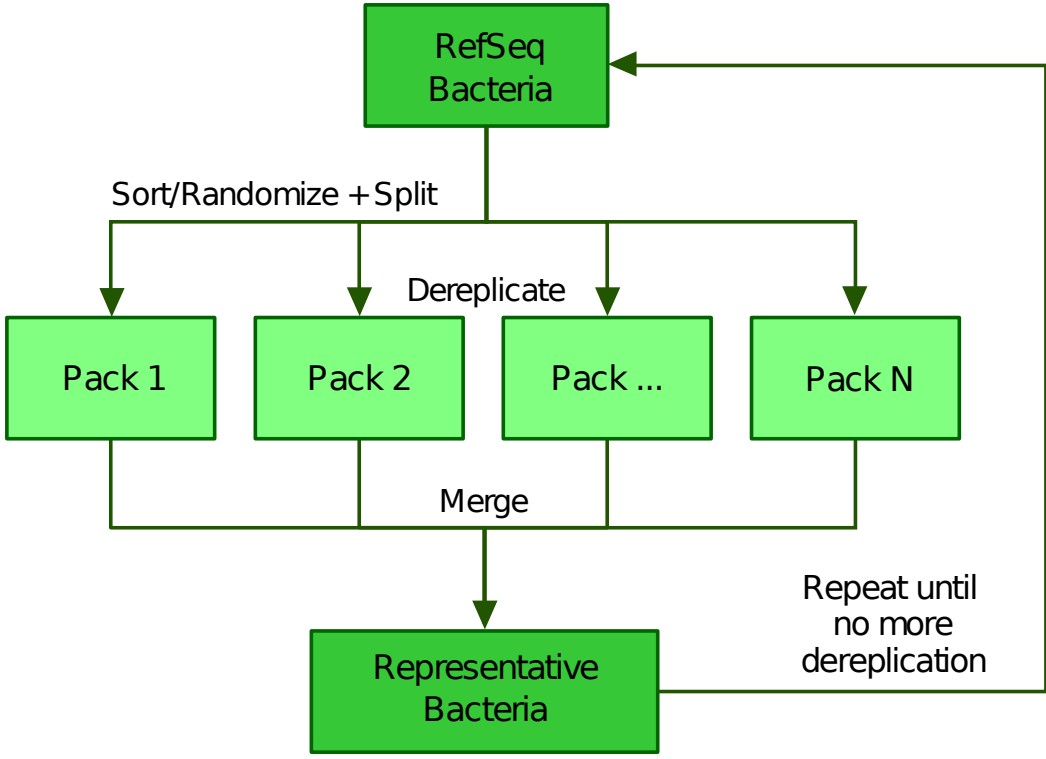

**Figure 2** **Illustration of the divide and conquer strategy of the dereplication phase.** From a list of Bacteria downloaded from RefSeq (or GenBank), TQMD either sorts (based on the NCBI taxonomic lineage of each genome) or randomizes the list and splits it into packs of a given size. This allows each pack to be separately dereplicated, especially in parallel. Then all resulting lists of representative genomes are merged back and TQMD decides if it can stop or must refeed the merged list for another round.

we used Scafos v1.30k (*Roure, Rodriguez-Ezpeleta & Philippe, 2007*) to create the nine different supermatrices, using the Minimal evolutionary distance as a criterion for choosing sequences, the threshold set at 25%, the maximal percent of missing sites for a "complete sequence" set to 10 and the maximum number of missing OTUs set to 25, except for Firmicutes (22). Finally, IQ-TREE (*Nguyen et al., 2015*; *Hoang et al., 2018*) was used to infer the phylogenomic tree associated with each supermatrix, using the LG4X model with ultrafast bootstraps. Trees were automatically annotated and colored using format-tree.pl (also from Bio::MUST::Core) and then visualised with iTOL v4 (*Letunic & Bork, 2019*). The whole pipeline, from the launch of TQMD to the tree produced by IQ-TREE required approximately 3 working days for the larger bacterial selection (Table 1, line B).

# RESULTS AND DISCUSSION

The TQMD workflow has two separate phases: a preparation phase (Fig. 1A) and a dereplication phase (Fig. 1B). The objective of the preparation phase is to compute the genome-specific data that will be needed during the dereplication phase. These operations are embarrassingly parallel and very easy to speed up. In contrast, the dereplication

**Table 1** **Details of TQMD runs and phylogenomic datasets built on eight different subsets of Bacteria.** For each dataset, TQMD was launched with the Jaccard Index as a distance, a pack size of 200, the loose clustering mode, and was allocated a maximum of 50 CPUs. Other parameters (direct or indirect strategy and distance threshold) are provided in the table, along with the total running time in CPU hours (h.CPU), the initial number of genomes (# starting), the number of representatives obtained (# repr.), the number of ribosomal protein alignments used in the supermatrix (# prot.), and the number of unambiguously aligned amino acids in the supermatrix (# AA). Further details (taxonomy and download links, Krona taxonomic plots, Forty-Two reports, supermatrices and trees) are available at https://doi.org/10.6084/m9.figshare.13238936.

| Label | Dataset | Strategy | Threshold | h.CPU | # starting | # repr. | # prot. | # AA |
|-------|---------|----------|-----------|-------|-----------|---------|---------|------|
| A | Bacteria (49) | indirect | 0.900 | 656 | 63,863 | 49 | 53 | 6338 |
| B | Bacteria (151) | indirect | 0.880 | 656 | 63,863 | 151 | 53 | 6187 |
| C | Actinobacteria | direct | 0.900 | 96 | 8859 | 20 | 51 | 6562 |
| D | Bacteroidetes | direct | 0.850 | 16 | 1225 | 37 | 49 | 6605 |
| E | Chlamydia | direct | 0.800 | 6 | 360 | 32 | 44 | 6131 |
| F | Cyanobacteria | direct | 0.800 | 8 | 428 | 46 | 48 | 6314 |
| G | Firmicutes | direct | 0.900 | 242 | 21,544 | 22 | 52 | 6536 |
| H | Proteobacteria | direct | 0.885 | 310 | 30,690 | 36 | 53 | 6471 |
| I | Archaea | direct | 0.850 | 8 | 432 | 86 | 57 | 7810 |

phase considers all genomes at once, with the aim of clustering similar genomes based on pairwise distances and selecting the best representative for each cluster. To achieve this in the presence of many genomes, TQMD resorts to a greedy iterative heuristic in which each round is parallelized through a divide-and-conquer approach. The two phases are interconnected by the means of a relational database (see 'Materials and Methods' for details). Hereafter, we study the effects of TQMD parameters and heuristics on its dereplication behavior, then we compare its performance to those of two similar solutions, dRep and Assembly Dereplicator and, finally, we provide some application examples in the field of prokaryotic phylogenomics.

## Analysis of TQMD behavior, parameters and heuristics

The dereplication phase is governed by a number of parameters and heuristics. One important issue is the inter-genome distance, which can either be based on the well-known Jaccard index (JI) or the identical genome fraction (IGF; see Materials and Methods for details). The latter was developed in an attempt to handle the comparison of genome pairs in which one is either partial or strongly reduced due to streamlining evolution or metagenomic source (*Cornet et al., 2018a*). Whatever the selected distance, genomes that are less distant than a user-specified threshold will end up in the same cluster. This distance threshold is thus the main "knob" for controlling the aggressivity of TQMD dereplication: the higher the threshold the tighter the clustering. Another point to consider are TQMD heuristics and their parameterization. Since TQMD is iterative, one can always decide to dereplicate genomes that are themselves representatives obtained in one or more previous runs. When trying to dereplicate very large and taxonomically broad genome sets, this raises the possibility to "guide" the dereplication by first clustering several phylum-wide subsets before merging the selected representatives in a single dataset to be dereplicated once more. This "indirect strategy" is to be contrasted with the "direct strategy", in which TQMD is left

dealing with the whole dataset from the very beginning. Regarding the divide-and-conquer algorithm operating during a single round, four parameters might be relevant: the pack size (e.g., 200 to 500), the clustering mode (loose or strict) and the dividing scheme (random or taxonomically-guided). Obviously, larger pack sizes require more time to be processed but are less likely to be affected by the impossibility to dereplicate two genomes that are in different packs. The clustering mode will also influence the number of pairwise comparisons required and thus the time necessary to cluster the genomes within a pack. Finally, in an attempt to balance such negative effects and the clustering speed, genome packs can either be composed at random (random sort) or by preferentially grouping taxonomically related organisms (taxonomic sort).

### Performance criteria

Before studying the behavior of TQMD under different sets of parameters and heuristics, one has to keep in mind that its aim is to generate dereplicated lists of genomes that maintain the phylogenetic diversity of the input genomes, especially at the highest levels of the prokaryotic taxonomy. Therefore, we identified two metrics of interest when examining TQMD output: (1) the number of phyla with at least one representative genome ("diversity") and (2) the taxonomic mixing amongst the clusters ("mixity"). The diversity can be put in perspective with the number of representatives using what we call a redundancy index, i.e., the number of representatives divided by the number of phyla, with the lower the better. Regarding the concept of taxonomic mixing, we use it when the group of genomes behind a representative genome is not taxonomically homogeneous at some specific taxonomic level. Since our objective is mostly to dereplicate at the phylum level, we checked the taxonomic mixing at the phylum level. For example, if within a group of Proteobacteria, one (or several) Firmicutes is present, then the group is considered "mixed".

### Iterative algorithm: dereplication kinetics

We first compared the results of the two distance metrics (JI or IGF) on the full set of RefSeq Bacteria passing our quality control (see Materials and Methods). To study the effect of the distance threshold used for dereplication, we selected two ranges of six values giving similar final numbers of representatives for the two metrics (JI: from 0.8 to 0.9; IGF: from 0.6 to 0.7). Figure 3 shows the dereplication kinetics observed when using a medium threshold (JI: 0.84; IGF: 0.66) and the direct strategy. The extreme efficacy (i.e., clustering ratio; see Materials and Methods) of the first round of dereplication is clear and subsequent rounds reach a plateau almost immediately. Whereas there is no notable difference between the two metrics in terms of kinetics, the height of the plateaus are not the same, with the IGF distance appearing greedier than the JI distance, especially when considering represented phyla rather than representative genomes.

### Iterative algorithm: effect of parameters and heuristics

While TQMD was designed to be run without manual intervention (direct strategy), it is also possible to funnel the process by feeding it taxonomically homogeneous subsets of representative genomes (indirect strategy). To contrast the two strategies, we first separated

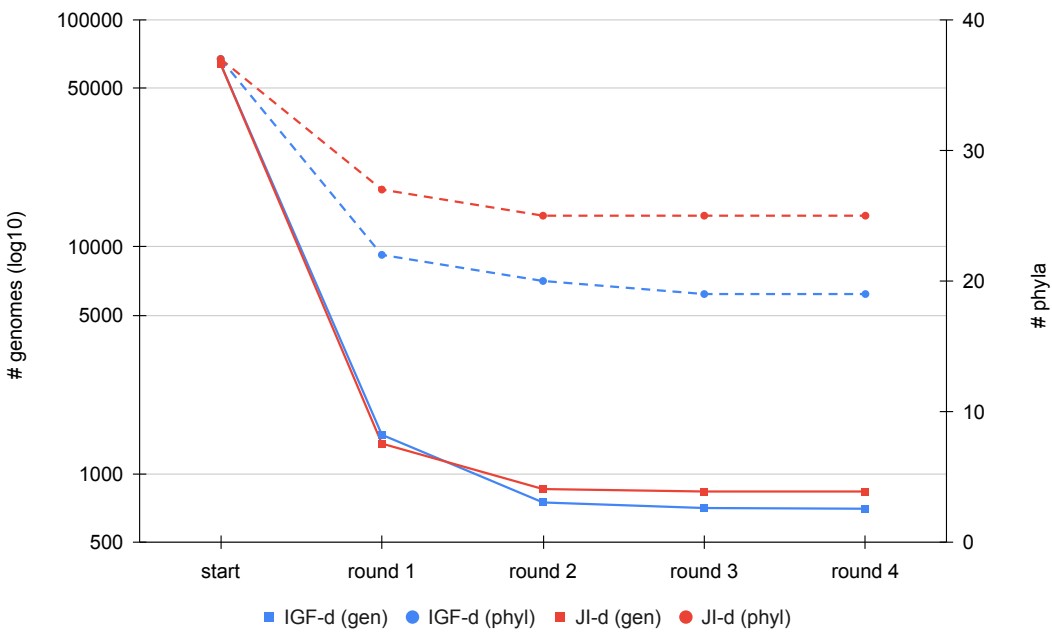

**Figure 3** **Comparison of the dereplication kinetics of TQMD when varying the distance metric.** Two runs were launched on all RefSeq Bacteria (63,836 genomes; 37 phyla) using the direct strategy, a pack size of 200 and the loose clustering mode, one with the Jaccard Index (JI-d, distance threshold of 0.84, red curves) and one with the Identical Genome Fraction (IGF-d, distance threshold of 0.66, blue curves). The left $Y$-axis shows the log10 of the number of remaining genomes (square dots and solids lines), whereas the right $Y$-axis shows the number of phyla for which at least one representative is still present at a given round of dereplication (round dots and dashed lines).

Bacteria into five groups corresponding to the four largest phyla in terms of numbers of genomes available in RefSeq and a fifth group with the rest of Bacteria: Proteobacteria (39,011 genomes), Firmicutes (26,972 genomes), Actinobacteria (10,248 genomes), Bacteroidetes (1,639 genomes), other bacteria (2,682 genomes).Then we dereplicated the four phyla separately using the JI and a distance threshold of 0.8. Finally, we pooled the representatives obtained through the four TQMD runs with the remaining Bacteria and launched a final run on this reconstructed list. For this final run, we tried the two metrics and the full range of thresholds. The results of this multidimensional comparison are provided in Table 2 and Fig. 4.

Starting with an initial number of bacterial phyla equal to 37, it appears that the two JI strategies are better than any IGF strategy in terms of diversity, since the former retain a higher number of represented phyla for a given number of representative genomes. For example, when ending with about 500 representatives, the JI distance preserves 22–24 phyla, whereas the IGF distance only retains 15–19 phyla. These numbers translate to redundancy index (RI) values of 25–20 (JI) and 31–23 (IGF), respectively (Table 2). With the IGF distance, the indirect strategy appears better at all thresholds, with a number of represented phyla systematically higher for a number of representatives systematically lower. This translates to, e.g., RI = 50 (IGF-i) vs 65 (IGF-d) with about 1550 representatives and RI = 30 (IGF-i) vs 33 (IGF-d) for about 720 representatives. In contrast, this is less

**Table 2** Comparison of the clustering properties when varying the distance metric, the distance threshold or the clustering strategy. Analyses were run on 63,863 RefSeq Bacteria using two different distance metrics, either based on the Jaccard Index (JI) or the Identical Genome Fraction (IGF), six different distance thresholds (from 0.8 to 0.9 and from 0.6 to 0.7, respectively), and two different clustering strategies, either direct (JI-D and IGF-D) or indirect (JI-i and IGF-i; see text for details). All pack sizes were 200 and the clustering mode was set to "loose". RI, Redundancy Index (# groups / # phyla).

### Jaccard Index (JI)

| Direct strategy (JI-d) | | | | | | | Indirect strategy (JI-i) | | | | | | |
|---|---|---|---|---|---|---|---|---|---|---|---|---|---|
| threshold | 0.80 | 0.82 | 0.84 | 0.86 | 0.88 | 0.90 | threshold | 0.80 | 0.82 | 0.84 | 0.86 | 0.88 | 0.90 |
| RI | 59 | 47 | 35 | 25 | 14 | 10 | RI | 54 | 46 | 35 | 20 | 12 | 4 |
| # phyla | 34 | 34 | 29 | 24 | 19 | 11 | # phyla | 34 | 31 | 25 | 22 | 13 | 11 |
| # groups | 2005 | 1589 | 1025 | 598 | 268 | 109 | # groups | 1845 | 1430 | 870 | 446 | 151 | 49 |
| —pure groups | 1992 | 1576 | 1009 | 587 | 261 | 106 | – pure groups | 1835 | 1416 | 853 | 434 | 149 | 45 |
| – singletons | 1201 | 904 | 557 | 325 | 143 | 56 | – singletons | 1727 | 818 | 488 | 242 | 88 | 24 |
| — mixed groups | 13 | 13 | 16 | 11 | 7 | 3 | – mixed groups | 10 | 14 | 17 | 12 | 2 | 4 |
| – paraphyletic | 0 | 0 | 0 | 0 | 0 | 0 | – paraphyletic | 0 | 1 | 0 | 0 | 0 | 0 |
| – super-phyla | 10 | 10 | 12 | 5 | 2 | 0 | – super-phyla | 10 | 13 | 9 | 7 | 0 | 1 |
| – polyphyletic | 3 | 3 | 4 | 6 | 5 | 3 | – polyphyletic | 0 | 0 | 8 | 5 | 2 | 3 |

### Identical Genome Fraction (IGF)

| Direct strategy (IGF-d) | | | | | | | Indirect strategy (IGF-i) | | | | | | |
|---|---|---|---|---|---|---|---|---|---|---|---|---|---|
| threshold | 0.60 | 0.62 | 0.64 | 0.66 | 0.68 | 0.70 | threshold | 0.60 | 0.62 | 0.64 | 0.66 | 0.68 | 0.70 |
| RI | 74 | 65 | 58 | 45 | 33 | 31 | RI | 50 | 55 | 44 | 30 | 23 | 11 |
| # phyla | 24 | 24 | 22 | 22 | 22 | 15 | # phyla | 31 | 25 | 24 | 24 | 19 | 16 |
| # groups | 1776 | 1548 | 1271 | 988 | 719 | 464 | # groups | 1536 | 1369 | 1061 | 715 | 440 | 176 |
| —pure groups | 1758 | 1530 | 1271 | 971 | 706 | 456 | – pure groups | 1514 | 1345 | 1042 | 701 | 426 | 167 |
| – singletons | 1094 | 939 | 755 | 587 | 419 | 260 | – singletons | 905 | 784 | 595 | 404 | 219 | 77 |
| —mixed groups | 18 | 18 | 19 | 17 | 13 | 8 | – mixed groups | 22 | 24 | 19 | 14 | 14 | 9 |
| – paraphyletic | 4 | 2 | 2 | 2 | 1 | 1 | – paraphyletic | 2 | 3 | 1 | 0 | 2 | 0 |
| – super-phyla | 11 | 11 | 13 | 10 | 4 | 1 | – super-phyla | 17 | 17 | 14 | 10 | 8 | 4 |
| – polyphyletic | 3 | 5 | 4 | 5 | 8 | 6 | – polyphyletic | 3 | 4 | 4 | 4 | 4 | 5 |

obvious with the JI distance, where the indirect strategy does not perform significantly better, the number of representatives also decreases but the number of represented phyla is also lower (or equal for the 0.9 threshold).

In the majority of the groups, the genome count per cluster is low with a significant proportion of singletons (i.e., only one representative genome, Table 2). However, in a few cases, large phyla (e.g., Proteobacteria, Firmicutes) gather into mixed groups that reach extreme genome counts and are visible as peaks in Fig. 4. Neither strategy changes this tendency but it is of notice that the JI distance with the indirect strategy is the combination leading to the lowest genome count per cluster and the lowest count of mixed groups (Table 2 and panel JI-i in Fig. 4), indicating a tendency to prevent the appearance of polyphyletic groups. When looking at the mixing (Table 2), it appears that unless at the highest thresholds, the mixity remains marginal in all strategies. To analyze the situation within the mixed groups, we separated them into three categories: (1) paraphyletic groups (only one case, Firmicutes and Tenericutes), (2) super-groups (e.g., FBC, PVC, Terrabacteria; see Fig. 5), and (3) polyphyletic groups. Since the TQMD objective is aggressive dereplication,

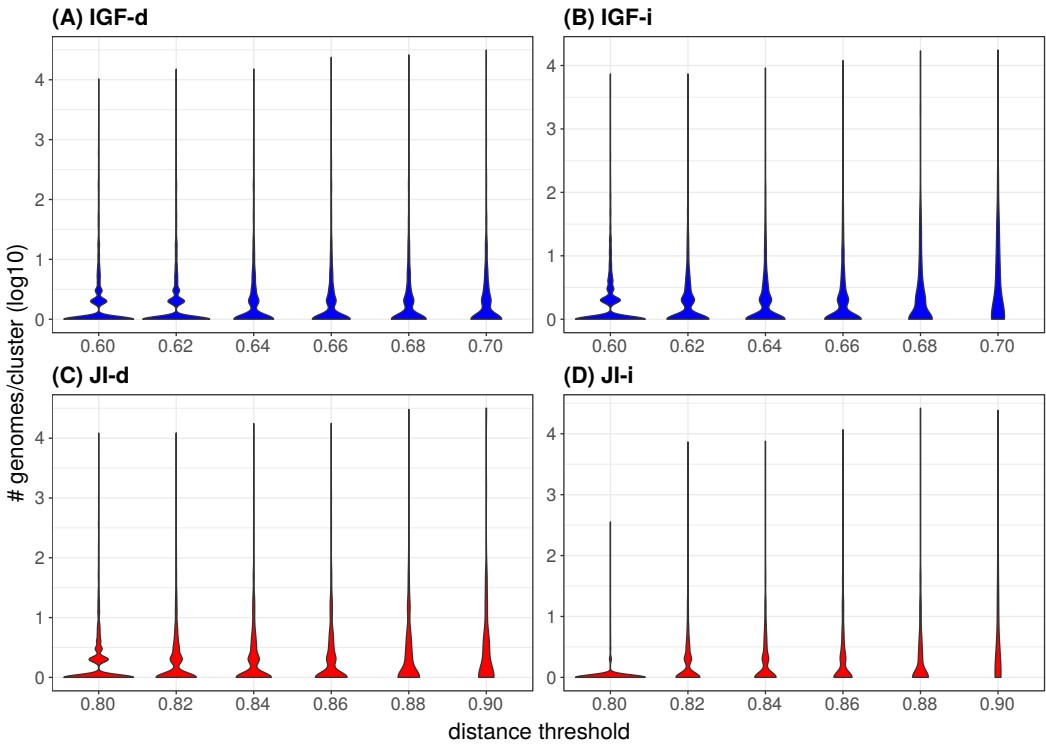

**Figure 4** **Distribution of the number of genomes per cluster when varying the distance metric, the distance threshold or the clustering strategy.** (A) IGF-d, (B) IGF-i, (C) JI-d, (D) JI-i. These violin plots are a companion to Table 3 and abbreviations are as in the latter table. The $Y$-axes are in log10 units and the violin plot width is proportional to the number of clusters containing the given number of genomes.

the first two types of mixing are not problematic. Indeed they show that TQMD works as intended by first regrouping similar genomes together before regrouping the more dissimilar genomes. This also confirms that multiple scales of genuine phylogenetic signal lie in the nucleotide $k$-mers used in TQMD (*Wen et al., 2014*; *Allman, Rhodes & Sullivant, 2017*).

Amongst polyphyletic groups, the "early" groups, i.e., those that appear at lower thresholds (0.8 for JI and 0.6 for IGF), are (1) Firmicutes/Tenericutes clustered with Thermotogae and other thermophilic bacteria and (2) Terrabacteria clustered with Synergistetes. Thermotogae are likely mixed with Firmicutes due to their chimeric nature, Firmicutes being one of the main gene contributors (through lateral gene transfer, LGT) to Thermotogae (*Nesbø et al., 2009*; *Gupta & Bhandari, 2011*). At higher thresholds, Thermotogae attract the other thermophilic bacteria, leading to the formation of a polyphyletic group. This result is a consequence of our single-linkage approach, which reveals to be a weakness when it comes to chimeric organisms that can bridge unrelated bacterial genomes. It might be possible to alleviate this effect by using the strict clustering mode (see below). Regarding the clustering of Synergistetes with other Terrabacteria, when only a few genomes were available, Synergistetes were dispersed within two other phyla, Deferribacteres and Firmicutes (*Jumas-Bilak, Roudiere & Marchandin, 2009*). Nowadays,

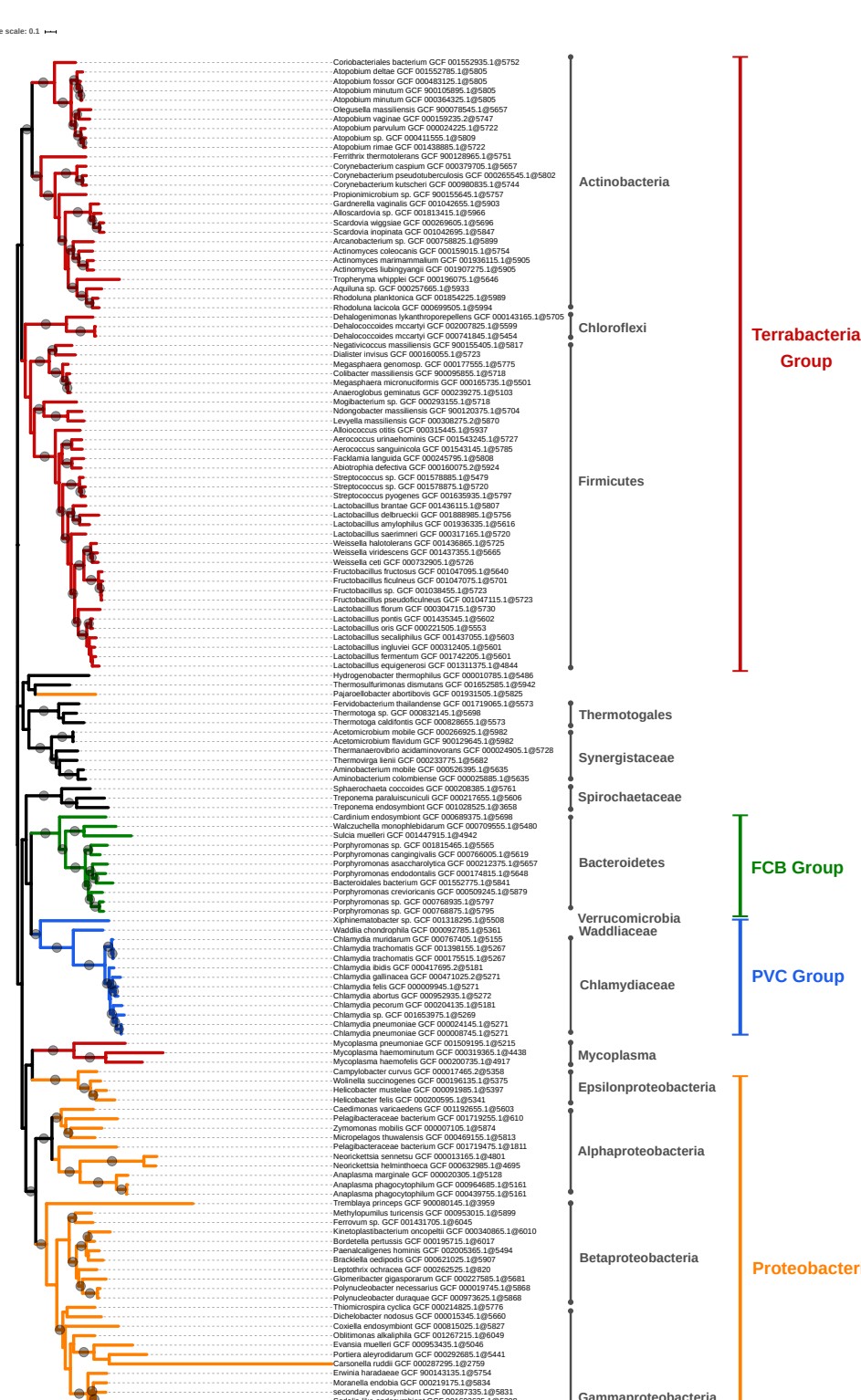

**Figure 5  Phylogenomic tree of the largest selection of Bacteria.** Tree inferred from a supermatrix of concatenated ribosomal proteins (Table 1B) under the LG4X model using IQ-TREE. Dots on branches indicate maximum bootstrap support values (100%).

Synergistetes form a monophyletic group that is sister to Deferribacteres (*Jumas-Bilak, Roudiere & Marchandin, 2009*). We hypothesize that conflicting (maybe artifactual) signals cause (at least some) Synergistetes to cluster with Firmicutes, and then to attract other Terrabacteria in a snow-ball effect due to single linkage. In other words, as the thresholds are increased, Thermotogae and Synergistetes serve as bridges between other bacterial phyla, creating or enlarging polyphyletic groups. This highlights that, just like alignment-based phylogeny, *k*-mer based approaches are also affected by chimeric organisms and LGT (*Daubin, Moran & Ochman, 2003*).

### Divide-and-conquer algorithm: effect of parameters and heuristics

With respect to the parallelization of TQMD, the pack size has an influence on the results, since every time the size is diminished, the number of representatives returned at the end increases, whatever the distance metric (Table S1). This can be explained easily. In each pack, there is a list of genomes, to which each genome is compared in turn until it finds a cluster to join or creates a new cluster on its own. For each group, the selected representative is the best genome to work with in downstream applications, but not the "centroid" genome for the cluster. This means that a representative can be in the "outskirt" of its cluster in terms of sequence, which makes it less able to attract other genomes in subsequent groups. On the opposite, the single-linkage approach of the loose mode helps to alleviate the outskirt effect by enabling a genome to join a cluster as soon as any genome of that cluster is within the specified distance threshold. Another way to solve this issue is by increasing the pack size yet at the cost of speed. For example, 25 genomes require approximately 30 min to be processed, while 200 genomes take 2 h and 500 genomes take several days, which corresponds to a quadratic complexity.

The clustering mode (either loose and strict) also affects the clustering results. In Table 3, when compared to the corresponding (upper-left) part of Table 2, the effect of the strict mode on the number of representatives is obvious. As expected, they are more numerous than in loose mode since it becomes more difficult to cluster genomes together. Yet, if this effect is noticeable at the lower distance thresholds, it is barely noticeable at the higher thresholds. A second effect is that the polyphyletic groups of mixed genomes appear later (i.e., at higher thresholds) in strict mode than in loose mode.

Finally, TQMD tries to speed up the dereplication process by assembling packs following a taxonomic sort of the genomes to dereplicate. This heuristic should improve the clustering ratio of each iteration by directly comparing genomes that are more likely to be similar, thereby greatly reducing the required number of rounds of the whole process. As expected, five independent runs launched on all RefSeq Bacteria using JI-d (Table 4) with genomes sorted randomly returned selections of 904 representatives (on average) in 17 to 18 rounds whereas, the same run with genomes sorted according to taxonomy returned 836 representatives in only four rounds. Similarly, five runs using IGF-d with genomes sorted randomly yielded 456 representatives (on average) in 9 to 10 rounds, in contrast to 702 representatives in four rounds by enabling the taxonomic sort. However, when dereplicating subsets corresponding to Proteobacteria, the random dividing scheme returned less representatives (124, worst result) than the taxonomic dividing scheme (165),

**Table 3  Effect of the strict clustering mode on the clustering properties when varying the distance threshold.** Analyses were run on 63,863 RefSeq Bacteria using the Jaccard Index and the direct strategy (JI-d) with six different distance thresholds (from 0.8 to 0.9). All pack sizes were 200. RI, Redundancy Index (# groups / # phyla). This table has to be compared to the upper-left quarter of Table 2.

| Thresholds | 0.80 | 0.82 | 0.84 | 0.86 | 0.88 | 0.90 |
|---|---|---|---|---|---|---|
| RI | 66 | 49 | 37 | 24 | 15 | 8 |
| # phyla | 34 | 33 | 28 | 26 | 20 | 14 |
| # groups | 2231 | 1609 | 1035 | 614 | 300 | 112 |
| - pure groups | 2220 | 1592 | 1021 | 598 | 283 | 104 |
| – singletons | 1289 | 875 | 551 | 328 | 149 | 52 |
| - mixed groups | 11 | 17 | 14 | 16 | 17 | 8 |
| – paraphyletic | 1 | 0 | 0 | 0 | 2 | 0 |
| – super phyla | 10 | 16 | 10 | 10 | 11 | 4 |
| – polyphyletic | 0 | 1 | 4 | 6 | 4 | 4 |

in approximately the same number of rounds (3 to 5). Similar results were observed with Firmicutes: 224 representatives using the random scheme (worst result) vs 333 representatives using the taxonomic scheme. These results suggest that the random sort can be useful while working with a taxonomically homogeneous subset of bacteria. In other cases, it should be avoided because a higher number of rounds translates to a longer computing time.

### A word about the genome source

In addition to RefSeq genomes, TQMD can also download and cluster GenBank genomes, along with (optional) custom genomes provided by the user. To test the effect of the source database, we studied the dereplication of RefSeq and GenBank Archaea (release 203), which have the advantage of combining a small number of genomes (941 and 4129 genomes, respectively) while featuring a lot of unclassified organisms, candidate phyla and metagenomic assemblies in GenBank (Table 5). Beyond the speed penalty due to sheer difference in the number of genomes, which influences the number of comparisons TQMD has to perform, switching to GenBank as the genome source also requires using canonical $k$-mers to account for the lesser assembly quality of many genomes (see Materials and Methods for details) and/or selecting Mash as the $k$-mer engine. Moreover, with GenBank, the diversity of representative genomes is expanded with candidate phyla, but at the cost of more unclassified genomes and also (meta)genomes of lesser assembly quality. Unclassified genomes are genomes without higher-level taxonomic taxa, which hinders the taxonomic sort heuristic and makes it harder for TQMD to dereplicate them (since they can start in packs distinct from those including the genomes they are the most similar to). Regarding genomes of lesser quality, some can act as a bridge between two clusters that should not be clustered together (as discussed above with the polyphyletic groups) if they are chimerical in any way (either genuinely or due to the mixing of different organisms). In the worst case, all genomes end up lumped together in a single large cluster (last row of Table 5). As our primary objective with TQMD was to provide high-quality representatives, we

**Table 4  Comparison of the number of rounds and final representatives when modifying the distance metric and/or the dividing scheme for parallel processing.** Five replicates of each combination were carried out for the random sort, whereas the taxonomic sort is deterministic. JI-based (direct) analyses were run using a distance threshold of 0.84, where IGF-based (direct) analyses used a threshold of 0.66. Pack size was 200 and the clustering mode was set to "loose".

| Dataset | dist./appr. | sort | # rounds | # repr. |
|---|---|---|---|---|
| Bacteria | JI-d | taxonomic | 4 | 836 |
| Bacteria | JI-d | random | 18 | 902 |
| Bacteria | JI-d | random | 17 | 903 |
| Bacteria | JI-d | random | 17 | 894 |
| Bacteria | JI-d | random | 18 | 915 |
| Bacteria | JI-d | random | 17 | 908 |
| Bacteria | IGF-d | taxonomic | 4 | 702 |
| Bacteria | IGF-d | random | 10 | 435 |
| Bacteria | IGF-d | random | 10 | 458 |
| Bacteria | IGF-d | random | 10 | 456 |
| Bacteria | IGF-d | random | 9 | 438 |
| Bacteria | IGF-d | random | 10 | 493 |
| Proteobacteria | IGF-d | taxonomic | 3 | 165 |
| Proteobacteria | IGF-d | random | 3 | 115 |
| Proteobacteria | IGF-d | random | 3 | 105 |
| Proteobacteria | IGF-d | random | 3 | 100 |
| Proteobacteria | IGF-d | random | 3 | 124 |
| Proteobacteria | IGF-d | random | 3 | 114 |
| Firmicutes | IGF-d | taxonomic | 4 | 333 |
| Firmicutes | IGF-d | random | 4 | 190 |
| Firmicutes | IGF-d | random | 5 | 212 |
| Firmicutes | IGF-d | random | 5 | 224 |
| Firmicutes | IGF-d | random | 4 | 194 |
| Firmicutes | IGF-d | random | 4 | 172 |

decided to focus this presentation on RefSeq, but Table 5 shows that TQMD also works with GenBank.

## Comparison with dRep, assembly-dereplicator and mash

When we began our work on TQMD in 2015, there was no published program for genome dereplication. Now two different software packages are available, dRep (*Olm et al., 2017*) and Assembly-Dereplicator, both built on top of Mash (*Ondov et al., 2016*). Mash itself was created to estimate the Jaccard distance (derived from the JI) within sets of genomes and metagenome-assembled genomes (MAGs) based on nucleotide $k$-mer counts (*Ondov et al., 2016*). dRep was designed especially for the dereplication of MAGs, whereas Assembly-Dereplicator (A-D) was designed for groups of bacteria which are sufficiently close relatives. A comparison of the working principles and features of dRep, A-D and TQMD is available in Table 6.

To compare TQMD to dRep (v2.2.3), we chose two different datasets from RefSeq (release 79), the phylum Bacteroidetes (1127 genomes) and the order Streptomycetales

**Table 5 Effect of the genome source (either RefSeq or GenBank) on clustering results using Archaea as a test case.** The runs carried out on GenBank Archaea used canonical k-mers. The JI runs used a distance threshold of 0.90 and the IGF runs a threshold of 0.80. The super-phyla are the Asgard group, the TACK group and the DPANN group. Unclassified genomes are genomes without a phylum in the NCBI Taxonomy. JI: Jaccard Index; IGF: Identical Genome Fraction.

| Source | # super-phyla | # phyla | # unclassified genomes | # genomes | Clustering mode |
|---|---|---|---|---|---|
| RefSeq | 3 | 7 | 0 | 941 | NA |
| GenBank | 3 | 24 | 265 | 4129 | NA |
| JI RefSeq | 2 | 6 | 0 | 46 | strict |
| JI RefSeq | 2 | 6 | 0 | 29 | loose |
| IGF RefSeq | 2 | 6 | 0 | 38 | strict |
| IGF RefSeq | 1 | 3 | 0 | 16 | loose |
| JI GenBank | 3 | 17 | 38 | 313 | strict |
| JI GenBank | 3 | 15 | 18 | 145 | loose |
| IGF GenBank | 2 | 10 | 6 | 34 | strict |
| IGF GenBank | 1 | 1 | 0 | 1 | loose |

(648 genomes; phylum Actinobacteria). Because of technical difficulties with the installation of dRep, we had to use a workstation less powerful than the grid computer used to run TQMD (see 'Materials and Methods'). That is why we did not use all the available bacterial genomes in these tests. Regarding Bacteroidetes, dRep required five hours (using 10 CPUs and default parameters) to select 835 genomes. With TQMD, we used a threshold of 0.6 on the JI to obtain comparable results. TQMD run lasted 10 h (on at most 6 CPUs) and selected 789 representative genomes, of which 707 were in common with those of dRep. Since our main objective is to maintain as much as possible the diversity when dereplicating, we verified how many species were retained after the dereplication. Before dereplication, we had 528 different species of Bacteroidetes; dRep produced a list covering 516 of these species, whereas TQMD produced a list of 517 species, of which 511 were in common (see Table 7 for details). With Streptomycetales, dRep (again using default values), selected 430 genomes out of 648 in approximately 12h30min using 20 CPUs. To emulate such a result with TQMD, we had to use a threshold of 0.4 and obtained 486 representatives (392 in common, of which 175 species) in about 10 h using at most 4 CPUs in parallel (details given in Table 6).

dRep is a less aggressive program than TQMD, which is unsurprising as the former is meant to be used on sets of MAGs and to dereplicate at the species level, while the latter is meant to be used on every completely sequenced prokaryotic genome available and to dereplicate at the phyla/class level. Moreover, from the very start, TQMD was designed with scalability in mind, so as to accommodate the ever growing number of sequenced genomes. In principle, dRep could be used aggressively like TQMD, by fine-tuning two different thresholds (primary and secondary clusters), but this would need dRep to allow the user to choose a different Mash $k$-mer size, which does not appear to be possible (for the average user). On the other hand, TQMD can be used to dereplicate down to the species level more easily (only one threshold to specify) but it would take a longer time to

**Table 6** Feature comparison between dRep, Assembly-Dereplicator (A-D) and TQMD.

| Feature | dRep | A-D | TQMD |
|---|---|---|---|
| main engine(s) | Mash + ANIm (or gANI) | Mash | JELLYFISH or Mash |
| other dependencies | CheckM (optional) | none | QUAST (optional), RNAmmer (optional), CD-HIT-EST (optional), Forty-Two (optional), CheckM (optional) |
| relational database | N | N | Y |
| genome source | custom | custom | RefSeq, GenBank, custom |
| taxonomic filters | N | N | Y (when downloading and clustering) |
| automatic genome download | N | N | Y |
| distance metric(s) | Mash distance (estimated JI) then ANI | Mash distance (estimated JI) | 1-JI (exact) or Mash distance (estimated JI) or 1-IGF (exact) |
| heuristic(s) | biphasic approach: Mash for fast and rough clustering followed by ANI for slow and accurate clustering | d-and-c strategy (serial) | iterative greedy algorithm (serial) + d-and-c strategy (parallel) |
| stop condition(s) | unspecified | first failure to dereplicate any serial batch | any of 3 possible cut-offs (number of rounds, number of representatives, clustering ratio) |
| d-and-c dividing scheme | unspecified | random | random or taxonomic |
| selection of representatives | formula based on genome size, assembly quality and contamination level (incl. strain heterogeneity) | assembly quality | formula based on genome size, assembly quality, annotation richness and contamination level (fully customisable with 30 possible metrics) |
| parameterization of representative selection | Y (parameter weights) | N | Y (simplified formula) |
| grid engine support | N | N | Y (SGE/OGE) (optional) |
| distribution | source (pip), conda, Galaxy | source | source (Bitbucket), Singularity container |
| CPU usage | fixed on launch | fixed on launch | specified as a maximum (decreases over time) |

Notes.
   JI, Jaccard Index; IGF, Identical Genome Fraction; ANI, average nucleotide identity; d-and-c, divide-and-conquer; SGE/OGE, Sun/Open Grid Engine; Y, present feature; N, absent feature.

**Table 7** Performance comparison between TQMD and dRep on two smaller datasets. # gen, starting number of genomes; # repr, final/common number of representative genomes; # spec, starting/final/common number of species; h.CPU, upper bound on CPU use (i.e., product of wall-clock time and number of CPUs). With TQMD, a distance threshold of 0.6 was used for Bacteroidetes and a threshold of 0.4 for Streptomycetales. In both cases, the pack size was 200, the clustering mode was set to "loose" and the taxonomic sort was selected.

| Dataset | Starting | | TQMD - JELLYFISH k12 | | | dRep | | | Intersection | |
|---|---|---|---|---|---|---|---|---|---|---|
| | # gen. | # spec. | # repr. | # spec. | h.CPU | # repr. | # spec. | h.CPU | # repr. | # spec. |
| Bacteroidetes | 1,127 | 528 | 789 | 517 | 60 | 835 | 516 | 50 | 707 | 511 |
| Streptomycetales | 648 | 220 | 486 | 207 | 40 | 430 | 189 | 250 | 392 | 175 |

finish since it would require a longer JELLYFISH $k$-mer size (see Material and Methods). In conclusion, the dRep and TQMD can do each other's work but become less efficient when trying to do so, thereby rather making them complementary: dRep to dereplicate at the species level and TQMD at phylum/class level. For intermediate taxonomic levels, it is up to the user to decide which one s/he prefers. It is of note that, except for the centrality metric of dRep, the five other metrics used by dRep are available amongst the 30 metrics offered by TQMD and can be used through its customisable ranking formula (see Materials and Methods).

A-D is a program that is more recent but, as of April 2021, not yet published; its last update dates from November 2019. Its main advantage is ease of use, since it is a simple (no-installation) script that only needs Mash as a prerequisite. A–D takes as input the path to a folder containing the genomes to be dereplicated and rearranges them randomly and separated into smaller packs (500 genomes per pack by default). The next step is the clustering of each pack serially using Mash. A–D stops as soon as it cannot dereplicate at least one genome from the current pack. However, at least in our hands, A-D revealed to be unstable and/or to perform poorly on our test datasets (see Supplementary Materials for details).

TQMD allows the use of two different $k$-mer engines, JELLYFISH and Mash. With JELLYFISH, TQMD can compute a distance that is based on the exact JI (or the exact IGF), whereas with Mash, it relies on a distance based on the estimate of the JI. From the user perspective, this means that a given distance threshold will not produce exactly the same results depending on the active $k$-mer engine. We compared the results and run times of JELLYFISH and Mash using RefSeq Cyanobacteria (release 203) (Table 8). At an equivalent $k$-mer size (12), Mash is indeed faster than JELLYFISH (in both strict and loose clustering modes) and produces a similar number of clusters. The speed benefit provided by Mash approximation allows the use of larger $k$-mers, as illustrated by the results of a run based on a $k$-mer size of 16, whereas such a setup would be computationally intractable with JELLYFISH. Therefore, the integration of Mash as a $k$-mer engine makes TQMD competitive even while dereplicating on lower taxonomic levels. Finally, the relationship between the distance threshold and the Jaccard distance is not straightforward, notably depending on the size ratio between the two genomes under comparison. To help with the selection of an appropriate threshold when using JELLYFISH, we produced Fig. S9 as a guideline. For Mash, we refer the reader to *Ondov et al. (2016)*, who provide similar information in their Fig. S3 (and Eq. (4)).

## Application example of TQMD

To check whether TQMD output was indeed useful in a practical context, we computed phylogenomic trees based on concatenations of ribosomal proteins sampled from selected representative genomes. We performed two runs on all RefSeq Bacteria (release 79; 63,863 genomes passing our prerequisites ; see Materials and Methods for details) using the indirect strategy and the JI, one at a distance threshold of 0.9 (Table 1, line A) and the other at 0.88 (Table 1, line B). The first run yielded a selection of 49 genomes while the second run retained 151 genomes. Seven additional runs using the direct strategy were

**Table 8  Comparison of run time for JELLYFISH/Mash and strict/loose modes.** All runs were carried out on RefSeq Cyanobacteria (918 genomes) using a distance threshold of 0.80 (JELLYFISH k12, 1-JI), 0.091 (Mash k12, Mash distance) and 0.069 (Mash k16, Mash distance). JI, Jaccard Index.

| k-mer engine | Time | | # representatives | |
|---|---|---|---|---|
| | **Strict** | **Loose** | **Strict** | **Loose** |
| Mash k12 | 0h56 | 1h44 | 73 | 49 |
| Mash k16 | 11h19 | 13h15 | 550 | 529 |
| JELLYFISH k12 | 3h14 | 7h30 | 73 | 52 |

carried out on the six largest bacterial phyla of RefSeq (in terms of numbers of organisms: Proteobacteria, Firmicutes, Actinobacteria, Bacteroidetes, Cyanobacteria and Chlamydia) and on Archaea. These phylum-wide selections contained about 20 to 50 genomes, each collectively representing the diversity of their respective phyla (Table 1, line C-H), whereas the archaeal selection contained 86 genomes (Table 1, line I). In this text, we only show and describe the larger phylogenomic tree of all Bacteria (Table 1, line B). The eight other trees are available as Figs. S1 to S8.

The larger bacterial tree (Fig. 5) results from an extremely aggressive selection (Table 1, B) but it still shows what we consider as the main groups of Bacteria (Proteobacteria, PVC, FBC, and "monoderm" phyla) and, after accounting for the idiosyncratic taxon names, most groups described by T. Cavalier-Smith (*Cavalier-smith & Chao, 2020*) are visible (with the exception of Eoglycobacteria and Hadobacteria, which were both absorbed in polyphyletic groups). Regarding the topology of the tree, all the organisms from the main super-phyla are generally regrouped in the same subtree, with some exceptions. These exceptions are the mycoplasma branch, which ends up within Proteobacteria, and *Pajaroellobacter abortibovis*, a proteobacterium that is separated from other Proteobacteria.

In Fig. 5, some genera and even species appear to be overrepresented in the selected genomes and form monophyletic subtrees within the tree. This is the case of *Lactobacillus*, for example, with 11 representatives (10 species). To investigate an eventual selection bias in TQMD, we launched two different TQMD runs using only the *Lactobacillus* genomes (841 which passed TQMD prerequisites). Both runs used the same values as the larger run for Bacteria (Table 1, B). The difference was the way of sorting the genomes before dividing them in packs, one used the taxonomic sort and the other the random sort. The run with the taxonomic sort yielded 19 *Lactobacillus* representatives (15 species), of which 10 in common with the larger run for Bacteria, whereas the random sort run yielded 21 representatives (16 species), of which 10 in common with the larger run for Bacteria and 16 with the taxonomic run. These results suggest that the taxonomic sort does not especially lead to a selection biased towards identically named genera or species, but that the representative genomes adequately sample the underlying phylogenetic diversity of the group. Along the same lines, dRep results for Bacteroidetes also show genomes of the same "species" not clustered together as in our Bacteroidetes tree (Table 6 and Fig. S3). This indicates that the genomes of such identically named organisms are actually quite different, thereby not reflecting a technical issue of TQMD or of dRep, but rather a genuine property of these genomes. Consequently, it is worth mentioning that a purely taxonomic (i.e., manual

based on NCBI Taxonomy) selection of representative genomes would have overlooked this genomic diversity, thereby reducing the relevance of the selection. In contrast, if the user is willing to accept a fixed number of representatives, a valuable alternative is to sample genomes from GTDB, since its taxonomy stems from ANI computations across RefSeq genomes, which is conceptually similar to what we dynamically do with TQMD. As for the current release (17/06/2020), GTDB features 111 "phyla" and 327 "classes" (*Parks et al., 2020*).

## CONCLUSION

TQMD is an efficient dereplication tool initially designed for the assembly of phylum-level datasets of representative prokaryotic genomes. It manages to maintain the taxonomic diversity of input genomes while being fast, owing to its aggressive dereplication heuristics, which makes it able to scale with the ever growing number of genome assemblies in public repositories, such as NCBI RefSeq and GenBank. At lower taxonomic levels, TQMD becomes slower, probably because it has to compare more genomes before finding pairs close enough to be clustered and dereplicated. However, the use of the "strict" mode for the clustering can at least partially offset this effect. To dereplicate at the lowest taxonomic levels (species or strains), a longer $k$-mer would be better suited. While this is computationally intractable with the JELLYFISH engine, the support of the faster Mash engine makes it possible. The development of the first version of TQMD is now finished and highly benefited from the input of *PeerJ* reviewers. Yet, it could be further improved by adding new distance metrics beyond JI and IGF, and/or by including additional metrics for the selection of representative genomes. And now, with the Singularity container, TQMD can even be run on a single-node computer without a scheduler, making it easier to install and use.

## ACKNOWLEDGEMENTS

The authors are grateful to Damien Sirjacobs for his support of the computing cluster and to Rosa Gago for her help with the design of the figures.

### Funding

Raphaël R. Léonard and Mick Van Vlierberghe were supported by FRIA fellowships of the Belgian National Fund for Scientific Research (F.R.S.-FNRS). Marie Leleu is supported by the French Agence Nationale de la Recherche (ANR, project MATHTEST). Frédéric Kerff is a Research Associate employed by the F.R.S.-FNRS. Computational resources were provided through two grants to DB (University of Liège "Crédit de démarrage 2012" SFRD-12/04; F.R.S.-FNRS "Crédit de recherche 2014" CDR J.0080.15). This work (and Luc Cornet) was also supported by a research grant to DB (no. B2/191/P2/BCCM GEN-ERA) funded by the Belgian Science Policy Office (BELSPO). The funders had no role in study design, data collection and analysis, decision to publish, or preparation of the manuscript.

## Grant Disclosures

The following grant information was disclosed by the authors:
Belgian National Fund for Scientific Research (F.R.S.-FNRS).
French Agence Nationale de la Recherche (ANR, project MATHTEST).
Belgian Science Policy Office (BELSPO): B2/191/P2/BCCM GEN-ERA.

## Competing Interests

The authors declare there are no competing interests.

## Author Contributions

- Raphaël R. Léonard performed the experiments, analyzed the data, prepared figures and/or tables, authored or reviewed drafts of the paper, developed the software ToRQuEMaDA, and approved the final draft.
- Marie Leleu performed the experiments, analyzed the data, prepared figures and/or tables, and approved the final draft.
- Mick Van Vlierberghe performed the experiments, prepared figures and/or tables, and approved the final draft.
- Luc Cornet performed the experiments, authored or reviewed drafts of the paper, developed the Singularity container, and approved the final draft.
- Frédéric Kerff and Denis Baurain conceived and designed the experiments, analyzed the data, authored or reviewed drafts of the paper, and approved the final draft.

## Data Availability

TQMD software is available at Bitbucket: https://bitbucket.org/phylogeno/tqmd.

The datasets are available at figshare: Léonard, Raphaël R.; Leleu, Marie; Van Vlierberghe, Mick; Cornet, Luc; Kerff, Frédéric; BAURAIN, Denis (2020): Datasets for Léonard et al. ToRQuEMaDA: Tool for Retrieving Queried Eubacteria, Metadata and Dereplicating Assemblies. figshare. Dataset. https://doi.org/10.6084/m9.figshare.13238936.v2.

## Supplemental Information

Supplemental information for this article can be found online at http://dx.doi.org/10.7717/peerj.11348#supplemental-information.

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
