# Peer review of "ToRQuEMaDA: tool for retrieving queried Eubacteria, metadata and dereplicating assemblies"

_PeerJ, doi:10.7717/peerj.11348_

## Round 0.1 · original submission · Major Revisions

The reviewers expressed very serious concerns regarding your manuscript. However, I believe that you will be able to answer all requirements and requests. Take your time.

Reviewer 1 ·

Basic reporting

The manuscript presents a tool (named TQMD) for removing redundant bacterial genomes from a set of assemblies. Specifically, the tool dereplicates genome assemblies by removing genomes for which there are sufficiently close relatives (defined by a distance threshold), resulting in a smaller set where the assemblies are more unique and maintain taxonomic diversity of input genomes. The topic is highly relevant to the genomics and metagenomics fields as the number of microbial genomes sequenced each year is expanding rapidly.

Overall, I think this is a nice piece of work, which can be potentially an important contribution in its field. I especially appreciated the care with which the representative genomes are selected by TQMD and the way the tool uses an internal database to store information about already analyzed genomes (only new genomes are added to the existing database). Nevertheless, I do have some concerns that I would like to see addressed by the authors.

1. Introduction needs more focus on the topic of genome dereplication and current tools designed for this purpose (e.g., dREP, Assembly-Dereplicator), rather than a general description of alignment-free methods for pairwise sequence comparison. I suggest the tuthors shorten the description of alignment-free methods and just provide the reader to the review (reference #1). For example, the paragraph about information theory-based methods (lines: 57-77) seems irrelevant to the study as all the tools for genome dereplication (including TQMD) are word-based (use k-mers) and do not use information theory concepts such as sequence complexity or compression. I suggest that the authors provide more justification for the study in the Introduction (specifically, by describing the existing tools for dereplication and presenting their advantages and disadvantages if possible).

2. I commend the Authors for their extensive 5-year work on the TQMD tool. However, throughout the manuscript, the authors describe TQMD as it is still unfinished. For example: "For now, we do not use all the metrics available in the TQMD database" (line: 232). In a similar manner, the authors state that 13 metrics of genome quality are retrieved by TQMD from the QUAST tool (line: 183), however, in the next sentence the authors stay that only three metrics were used (line: 192, "Among these metrics, we eventually decided to take into account...")

Experimental design

3. My main concern, which is also confirmed by the Authors, is that TQMD is only optimized for dereplicating genomes at high taxonomic levels (i.e., class and phylum) but not at lower taxonomic levels such as family, genus, and species. In NCBI RefSeq or GenBank, more than half of all available bacterial genomes belong to only several species (e.g., Escherichia coli, Staphyloccocus aureus, Bacillus subtilis), each present in thousands of almost identical copies. According to GtDB, which is an excellent curated resource for bacterial taxonomy built upon the latest RefSeq database (v. 95), there are 191,527 bacterial genomes encompassing about 16,000 representative species belonging to 111 phyla. Given the number of genomes belonging to one species is many times larger than the number of phyla there is a pressing need for efficient dereplication tools at the lowest taxonomic levels. As the authors admit in the manuscript, at low taxonomic levels TQMD does not work as accurately and efficient as the other two tools, dREP and Assembly-Dereplicator. It is clear to me that this low performance of TQMD is a result of using relatively short k-mers (~12 nucleotide), which are the upper bound to calculate the exact distance metrics (JI and IGF) in feasible time. However, I think TQMD might be improved to be at least as accurate as dREP at low taxonomic ranks (I elaborate my suggestions in the next two comments).

4. Calculating the exact JI and IGF distances in TQMD limits the size of k-mers that can be used. The authors are aware of this issue and concluded that "replacing Jellyfish by Mash would be interesting to compare the efficiency with the current version but this would imply dropping the IGF distance (since Mash only implements JI) and would require Mash to accept using different k-mers sizes". I think that the Authors can easily replace Jellyfish by Mash, preserving both JI and IGF distances. In addition to JI, Mash also outputs the number of shared k-mers in a sample (by default, the sample size is 1000, which is under the "-S (sketch size)" parameter in Mash). The authors could use this number to estimate IGF by slightly modifying the equation from line 256 to the following one: IGF(A,B) = |A ∩ B| / min(length of genome A, length of genome B). In this way, both the metrics would be approximated, allowing the selection of longer k-mers, thus making the TQMD tool also appropriate for low taxonomic levels. Of course, I am not forcing the Authors to do this, but I would appreciate the Authors to comment on this idea.

5. TQMD analyzes all 12-mers from one strand of genome sequence assuming the input genomes have the same strand orientation. I suggest the authors to use the "--canonical" parameter in Jellyfish to count k-mers on both strands and perform JI and IGF calculations on canonical k-mers only. This would also ease the calculations since the number of all possible 12-mers is 4^12, the number of canonical ones will be somewhere about (4^12)/2.

6. In line: 161-163 the authors state that Jellyfish crashes when using a size below 11 nucleotides. It is a known bug in Jellyfish v. 1.X, which was also confirmed by developers of Jellyfish. It is recommended to use Jellyfish in version >= 2.

7. When dereplicating genomes people usually want to cluster sequences that are at least XX% identical. For example, when clustering bacterial genomes belonging to the same species many studies use an average nucleotide identity (ANI) of 95% as a threshold. It would be useful to explain to readers how to set proper IJ or IGF thresholds in TQMD to obtain a desired sequence identity threshold for the clustering.

Validity of the findings

no comment

Additional comments

I very like the idea that representative genomes are expected to be fully sequenced, correctly assembled, richly annotated, and devoid of contaminating sequences.

Reviewer 2 ·

Basic reporting

no comment

Experimental design

The analysis of Assembly Dereplicator is problematic.

Validity of the findings

no comment

Additional comments

Léonard et al. propose a method for dereplicating genomic datasets. There is a need to dereplicate genomes given the large and rapidly growing number of available genome assemblies. However, the proposed method is computationally intensive and it is unclear that dereplication at higher taxonomic ranks is a problem. A common approach to building phylum or class level data sets is to manually select a single genome from each of these taxonomic groups. This ensure representation across known diversity and a representative from each named lineage. A tool to select a suitable representative for each higher-level taxon (i.e. phylum or class) would be useful, but this decision can be based purely on information pertaining to the quality of the assembly and available annotations. Generally speaking, assignment into different phyla or classes is sufficient to indicate genomes are diverse and that their incorporation into a dereplicated data set is warranted.

Major:
• As of this review (Dec. 1, 2020), TQMD did not work for me. I downloaded the source code, extracted the tarball, and ran “per Makefile.PL, “make”, and then “make test”. This failed with the message below. The README file on BitBucket would benefit from indicating the minimum Perl version required and all dependencies. Ideally, the installation script would install all required Perl modules. I am not familiar with Perl and was not able to install the software. A Conda package would be welcomed.
# Failed test 'use TQMD;'
# at t/00-load.t line 10.
# Tried to use 'TQMD'.
# Error: Can't locate Modern/Perl.pm in @INC (you may need to install the Modern::Perl module)
• The Introduction relies heavily on the prior work on Zielezinski et al. which was published in 2017. Can the authors comment on additional methods and software tools which may have been developed since 2017? Along these lines, the Introduction would benefit from introducing dRep and Assembly-Dereplicator in order to clarify why a comparison against these two tools is sufficient to establish the benefits and need for TQMD.
• Does TQMD only work on systems with SGE/OGE? This is restrictive as many labs do not run HPCs with this scheduler. I appreciate supporting multiple schedulers is not practical, but I would expect TQMD to be able to be run on a machine without requiring a scheduler. For example, we make common use of 64 CPU machines with >500 GB of RAM that are run without a scheduler.
• TQMD would prove useful to a much wider audience if there was also an option to work with genomes in NCBI’s GenBank assemblies instead of only RefSeq assemblies. My understanding is that RefSeq does not contain any MAGs or SAGs. Currently GenBank contains an additional 100,000+ genomes relative to RefSeq. Admittedly, these genomes are often of lower quality than those in RefSeq with many being metagenome-assembled genomes (MAGs) or single-cell genomes (SAGs). However, these genomes also span a substantial amount of diversity missed by the higher quality isolate genomes in RefSeq which is critical for many downstream analyses.
• Line 224: It is indicated that genome contamination will be estimated with the program Forty-Two. CheckM and BUSCO are established programs for calculating contamination which should give more accurate estimates than those based on just 42 ribosomal proteins. However, my major concern here is that it is indicated that Forty-Two “is not yet part of the automatic preparation pipeline”. How does TMQD obtain this information? This seems like an essential statistic, especially if MAGs are to be considered, and believe TMQD so incorporate this before it is made available to the wider research community.
• The global ranking of genomes used by TMQD is based determined using an “equal-weight sum-of-ranks approach”. It would be helpful to readers and myself if this statistical approach could be explained given that the ranking of genomes in a critical step. I am also unclear on what metric 5 is “number.enriched.alis / 90”.
• I have no trouble following the explanation of how the dereplication strategy works, but do not find the workflow given in Figure 1B clear. The flowchart makes no mention of genomes being processed in a specific order or clustering being performed in a greedy fashion based on specific threshold.
• TMQD dereplicates by comparing the current query genome to all genomes currently in clusters. It would substantially reduce the amount of computation if the query genome was only compared to the representative of each species cluster. This has the benefit that the resulting clusters are genomes that will guaranteed to be similar to the selected representative. Is there a disadvantage to this more computationally efficient dereplication strategy given the purpose of TMQD? For many purposes, it seems undesirable to have cluster where genomes may only be connected via a single linkage (i.e. genome A is similar to genome B is similar to … is similar to genome Z, but genome Z is highly divergent from genome A).
• I am unclear on the exact methodology used in the “Phylogenomic analyses” section. Earlier in the manuscript it is indicated there are 155,519 genomes as of RefSeq 202. However, the dereplication of the full bacterial sets appears to have started with only 63,863 genomes (datasets A and B in Table 1).
• The dereplication of the bacterial sets results in 49 and 151 representative genomes for a threshold of 0.1 and 0.12, respectively (Table 1). According to the latest version of the GTDB (https://gtdb.ecogenomic.org/) there are >300 bacterial classes. Can the authors comment on the benefits of TMQD as opposed to a taxonomy driven dereplication strategy where the best representative of each class is selected? This would seem to be far less computationally intensive and ensure a tree contains representatives across the current known diversity of the bacterial tree.
• The comparison to Assembly-Dereplicator starting on Line 535 is problematic. It appears that either Assembly-Dereplicator has a fundamental flaw in which case the comparison is not informative, or the program is not being run correctly. At a Mash distance threshold of 0.5 (~50% ANI), a substantial amount of dereplication should occur. Either way it is unclear what benefit providing that analysis has unless this issue can be addressed.

Minor:
• Is the software called TQMD or ToRQuEMaDA? I strongly prefer the former. It is unclear why the abbreviation ToRQuEMaDA requires its own abbreviation, and it is confusing that the title of the manuscript uses a different abbreviation than the Abstract.
• The Abstract has the phrase “…high taxonomic level…”. I think this should be “higher taxonomic levels”.
• The README file on the BitBucket page would benefit from removing the standard template sentences regarding the purpose of a README file and information about CPAN. It would be beneficial to make it clear that TQMD is a Perl program and thus requires Perl (certain version?).
• I do not believe that the Proteobacteria and Firmicutes are necessarily “hyper-sampled”. Perhaps these phyla just contain substantial diversity compared to other phyla.
• Line 222: I believe the clause “which could rule out some metagenomes” should be “which could rule out some metagenome-assembled genomes”?
• dRep and Assembly Dereplicator should be referenced when first mentioned.
• Line 220: it is indicated that “the presence of at least one SSU rRNA 16S predicted by RNAmmer is a requirement for the genome to be selected”. Is this true? Where is the presence of 16S rRNA genes used as a criterion by TQMD?
• Line 341: I believe the phrase “inter-distance genome” should be “inter-genome distance”.

---

## Round 0.2 · accepted · Accept

Both reviewers are fully satisfied now with your corrected version. I support their suggestion to accept your submission as it is.

Reviewer 1 ·

Basic reporting

-

Experimental design

-

Validity of the findings

-

Additional comments

I congratulate the authors for this very nice and thorough revision. I am delighted to recommend the manuscript for publication in PeerJ.

Reviewer 2 ·

Basic reporting

no comment

Experimental design

no comment

Validity of the findings

no comment

Additional comments

I thank the authors for their revisions and well articulated responses to my previous concerns.